# Leveraging Endo- and Exo-Temporal Regularization for Black-box Video Domain Adaptation

**Yuecong Xu**\*                                                                                     *xuyu0014@e.ntu.edu.sg*
*Department of Electrical and Computer Engineering*
*National University of Singapore*

**Jianfei Yang**\*                                                                                    *yang0478@e.ntu.edu.sg*
*School of Mechanical and Aerospace Engineering*
*Nanyang Technological University, Singapore*

**Haozhi Cao**                                                                                      *haozhi002@e.ntu.edu.sg*
*School of Electrical and Electronic Engineering*
*Nanyang Technological University, Singapore*

**Min Wu, Xiaoli Li**                                                                     *{wumin, xlli}@i2r.a-star.edu.sg*
*Institute for Infocomm Research*
*Agency for Science, Technology and Research (A\*STAR), Singapore*

**Lihua Xie**                                                                                          *elhxie@ntu.edu.sg*
*School of Electrical and Electronic Engineering*
*Nanyang Technological University, Singapore*
*IEEE Fellow*

**Zhenghua Chen**†                                                                                   *chen0832@e.ntu.edu.sg*
*Institute for Infocomm Research and Centre for Frontier AI Research*
*Agency for Science, Technology and Research (A\*STAR), Singapore*

**Reviewed on OpenReview:** *https://openreview.net/forum?id=icoP08mrQJ*

## Abstract

To enable video models to be applied seamlessly across video tasks in different environments, various Video Unsupervised Domain Adaptation (VUDA) methods have been proposed to improve the robustness and transferability of video models. Despite improvements made in model robustness, these VUDA methods require access to both source data and source model parameters for adaptation, raising serious data privacy and model portability issues. To cope with the above concerns, this paper firstly formulates Black-box Video Domain Adaptation (BVDA) as a more realistic yet challenging scenario where the source video model is provided only as a black-box predictor. While a few methods for Black-box Domain Adaptation (BDA) are proposed in the image domain, these methods cannot apply to the video domain since video modality has more complicated temporal features that are harder to align. To address BVDA, we propose a novel Endo and eXo-TEmporal Regularized Network (EXTERN) by applying mask-to-mix strategies and video-tailored regularizations. They are the endo-temporal regularization and exo-temporal regularization, which are performed across both clip and temporal features, while distilling knowledge from the predictions obtained from the black-box predictor. Empirical results demonstrate the state-of-the-art performance of EXTERN across various cross-domain closed-set and partial-set action recognition benchmarks, which even surpasses most existing video domain adaptation methods with source data accessibility. Code will be available at `https://xuyu0010.github.io/b2vda.html`.

---

\*Equal contribution.
†Corresponding author.

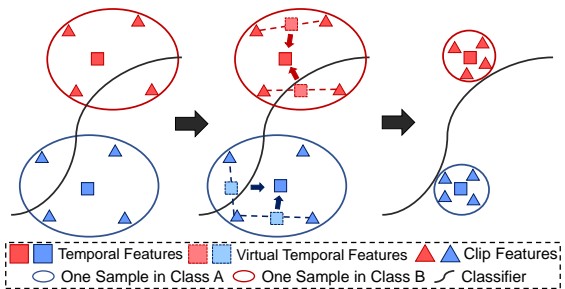

Figure 1: Clip features of target videos may be scattered, where clips from the same video may be separated by the decision boundary, resulting in differed label predictions. Such clip features violate both the cluster assumption and the *masked temporal hypothesis*. We augment the target video domain with virtual temporal features through a novel mask-to-mix strategy and apply endo-temporal regularization. The resulting temporal features are more discriminative and comply with both the cluster assumption and the *masked temporal hypothesis*.

# 1 Introduction

Video Unsupervised Domain Adaptation (VUDA) (Chen et al., 2019a; Xu et al., 2021a; 2022a) aims to transfer knowledge from a labeled source video domain to an unlabeled target video domain. It has wide applications in scenarios where massive labeled video data may not be available. Despite their effectiveness in improving the robustness of video models (Xu et al., 2022b), current VUDA methods require access to the source video data which contains personal and private information, raising serious concerns about data privacy and model portability (Liang et al., 2020a; 2022). The Source-Free Video Domain Adaptation (Xu et al., 2022c; Tian et al., 2021) is subsequently formulated to learn a target model without access to source data, yet it still relies on the well-trained source model parameters which allow generative models to recover source videos (Goodfellow et al., 2014; Creswell & Bharath, 2018; Luan et al., 2021; Wang et al., 2021).

Privacy-preserving is vital in applying action recognition models to real-world applications such as in smart hospitals and security surveillance where action recognition models are leveraged for anomaly behavior recognition (Sultani et al., 2018; Said et al., 2021; Zhou et al., 2019; Zhong et al., 2022). In these cases, current VUDA methods are totally inapplicable when sharing models across organizations due to their violation of privacy-related regulations, such as the European GDPR (Goddard, 2017) and Singaporean PDPA (Chik, 2013). To further cope with the video privacy issue, we formulate and study a more realistic and challenging video domain adaptation scenario termed the ***Black-box Video Domain Adaptation*** (BVDA) where the source video model is provided for adaptation only as a black-box predictor (e.g., API service). In privacy-concerned scenarios, BVDA helps to derive an accurate model in the target domain without access to both the parameters and data in the source domain.

Without access to source data and model, existing VUDA methods that aim at enhancing transferability through statistical alignment (e.g., TAMAN (Xu et al., 2023)) or adversarial alignment (e.g., TA³N (Chen et al., 2019a) and SAVA (Choi et al., 2020)) are not applicable. There have been a few recent research efforts (Liang et al., 2021; Yang et al., 2022) aiming at Black-box Domain Adaptation for images. One representative work is DINE (Liang et al., 2022), where target features are extracted by obtaining pseudo-labels from the black-box predictors while applying structural regularizations (Viola & Wells III, 1997; Verma et al., 2019) that encourage better model discriminability and generalization ability. However, structural regularizations of DINE are tailored for images that only contain spatial features. In comparison, characterized by the multi-modality nature, videos consist of spatial features and additional temporal information. This result in additional challenges in aligning temporal features. As a result, solutions for images such as DINE cannot show significant improvements for the BVDA task. Previous studies (Chen et al., 2019b; Yang et al., 2020a; Kundu et al., 2022; Huang et al., 2022) prove that improving discriminability would benefit the effectiveness of domain adaptation. Inspired by these studies, we propose to improve the discriminability of temporal features to tackle BVDA effectively when neither source data nor source model is accessible.

One common strategy to extract video temporal features is to split longer videos into a series of shorter clips. Therefore, temporal features can be constructed explicitly with the series of clip features (Wang et al., 2018; Zhou et al., 2018). Meanwhile, humans are capable of recognizing actions correctly even with only representative clips from videos (Isik et al., 2018). Intuitively, **if the target model is able to perform similarly to human perception and obtain discriminative features with consistent predictions given only partial clip information, the representational capacity of the target model and the discriminability of the extracted target temporal features could be improved significantly even without knowledge from the source domain**. We term the above hypothesis as the *masked-temporal hypothesis* as this hypothesis depicts the ideal characteristics of features obtained after certain clips are masked out. Our proposed method is built upon this hypothesis.

To this end, we propose the **E**ndo and e**X**o-**TE**mporal **R**egularized **N**etwork (**EXTERN**) to address the BVDA task. EXTERN extracts robust temporal features in a self-supervised manner by applying both the *endo-temporal regularization* and the *exo-temporal regularization* while distilling knowledge from the predictions obtained from the source predictor. Specifically, the endo-temporal regularization is designed to improve the discriminability of clip features by regularizing the behavior of clip features obtained **within the same target video**. This drives the clip features towards complying with the cluster assumption (Rigollet, 2007; Xiao et al., 2023) and the *masked-temporal hypothesis* as presented in Fig. 1. The endo-temporal regularization is achieved by augmenting the target video domain with virtual temporal features through a novel **mask-to-mix** strategy over clip features corresponding to the same video.

Meanwhile, the exo-temporal regularization is designed to drive the proposed model to extract more stable and discriminable temporal features, characterized by being linearly smooth in-between features obtained from **different target videos**. The exo-temporal regularization is achieved by augmenting the target video domain with interpolated temporal features. It is remarkable that our EXTERN achieves outstanding results, outperforming most existing VUDA methods that require source data and models. This demonstrates that training the target model from scratch may help overcome the negative effect of domain shift, paving a new way for tackling VUDA.

In summary, our contributions are threefold. (i) We formulated a realistic and more challenging task, *Black-box Video Domain Adaptation* (BVDA). To the best of our knowledge, this is the first work to address black-box domain adaptation for privacy-preserving and portable video model transfer. (ii) We propose EXTERN to address BVDA, which enhances discriminative temporal feature extraction through an endo-temporal regularization using a mask-to-mix strategy along with an exo-temporal regularization, driving clip features towards complying with the *masked-temporal hypothesis*. (iii) Extensive experiments show the efficacy of EXTERN, achieving state-of-the-art performances across cross-domain action recognition benchmarks under closed-set and partial-set video domain adaptation settings. EXTERN achieves remarkable 7.7% and 12.4% average improvements under the closed-set and the partial-set settings respectively, even outperforming most existing adaptation methods with access to source data.

## 2 Related Work

**(Video) Unsupervised Domain Adaptation ((V)UDA).** UDA and VUDA aim to distill shared knowledge across a labeled source domain and an unlabeled target domain, which improve the robustness and transferability of deep learning models (Zhao et al., 2020). (V)UDA methods could be broadly categorized into four categories: i) reconstruction-based methods (Ghifary et al., 2016; Yang et al., 2020b; Wei et al., 2022), where domain-invariant features are extracted by encoders trained with data-reconstruction objectives; ii) discrepancy-based methods (Saito et al., 2018; Xu et al., 2019; YUAN et al., 2022; Zhang et al., 2023), where domain alignment is achieved by applying metric learning approaches, optimized with metric-based objectives such as MDD (Zhang et al., 2019), CORAL (Sun et al., 2016), and MMD (Long et al., 2015); iii) adversarial-based methods (Tzeng et al., 2017; Xu et al., 2022b; Levi et al., 2022), where methods leverage additional domain discriminators along with feature generators, trained jointly in an adversarial manner (Huang et al., 2011) by minimizing adversarial losses (Ganin & Lempitsky, 2015); and iv) semantic-based methods (Choi et al., 2020; Kim et al., 2021; Sahoo et al., 2021; Song et al., 2021; Du et al., 2023), where domain-invariant features are obtained subject to certain semantic constraints by leveraging

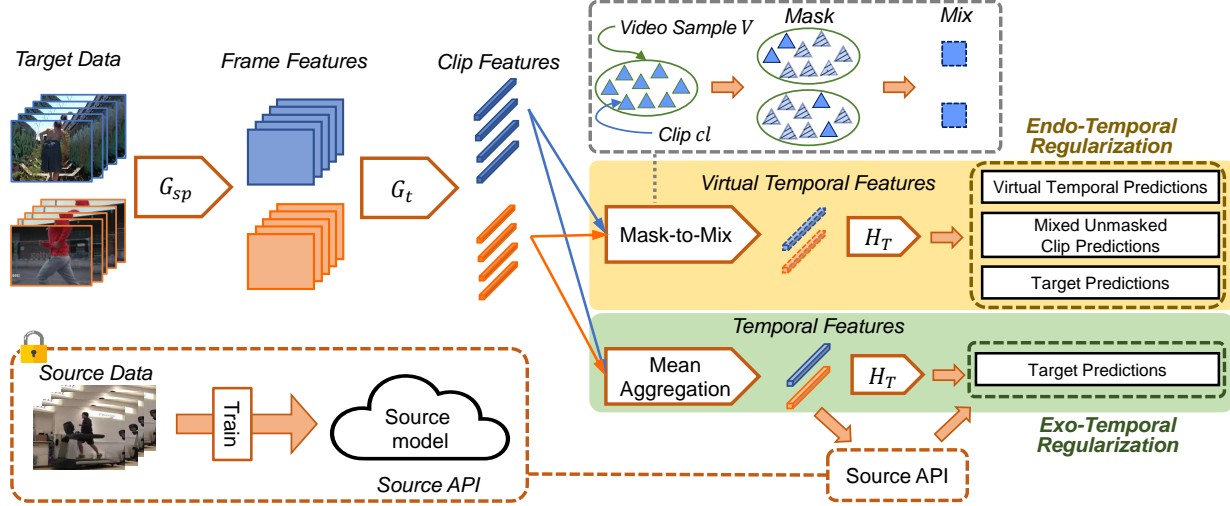

Figure 2: An overview of the proposed EXTERN. EXTERN extracts knowledge from the black-box source predictor (i.e., Source API) through a distillation process. EXTERN further extracts temporal features in a self-supervised manner by applying both the *endo-temporal regularization* and *exo-temporal regularization*. To apply the *endo-temporal regularization*, the virtual temporal features are constructed with a mask-to-mix strategy.

auxiliary tasks such as contrastive learning (Chen et al., 2020) and clip order prediction (Choi et al., 2020). Compared to UDA research which is primarily focused on image-based tasks, VUDA research lags behind owing to the challenges brought by aligning temporal features. Despite the challenges, there has been a substantial increase in research for VUDA, backed by the introduction of various cross-domain closed-set or partial-set video datasets (Chen et al., 2019a; Xu et al., 2021a; 2023). Regardless of the improvements in video model robustness and transferability, VUDA approaches all require access to both source data and source model parameters during adaptation, which could raise serious privacy concerns given the amount of private information of the subjects and scenes contained in videos.

**Black-box Domain Adaptation (BDA).** With the increased importance of data privacy with concerns of possible data leakage through white-box attack given model parameters (Zhang et al., 2020), there have been a few research that explores BDA with images. BDA enables image models to be adapted to the unlabeled target domain without either the source data or the source model parameters, with the source model provided only as a black-box predictor (Fang et al., 2022; Yu et al., 2023). Under such settings, BBSE (Lipton et al., 2018) focused on black-box label shift, and requires a hold-out source set for class confusion matrix estimation, which is commonly unavailable. More recently, LNL (Zhang et al., 2021) proposed to tackle BDA by an iterative noisy learning approach via pseudo labels that are refined with KL divergence, while DINE (Liang et al., 2022) leveraged knowledge distillation with information maximization and structural regularizations. Despite the above recent advances, BVDA has never been explored, which is more challenging as temporal features must also be aligned. We propose to tackle BVDA by applying a temporal feature tailored endo-temporal regularization leveraging a mask-to-mix strategy, along with exo-temporal regularization.

## 3 Methodology

For **Black-box Video Domain Adaptation** (BVDA), we only have access to a black-box video predictor $H_S$ (i.e., the constraint Source API) which is trained from the labeled source video domain $\mathcal{D}_S = \{(V_{iS}, y_{iS})\}_{i=1}^{n_S}$ containing $n_S$ videos, with $V_{iS} \in \mathcal{V}_S, y_{iS} \in \mathcal{Y}_S$, where $\mathcal{V}_S$ is the source input distribution and $\mathcal{Y}_S$ the source label space. We are also given an unlabeled target video domain $\mathcal{D}_T = \{V_{iT}\}_{i=1}^{n_T}$

with $n_T$ i.i.d. videos $V_{iT} \in \mathcal{V}_T$ where $\mathcal{V}_T$ is the target input distribution. Additionally, we assume that source and target video domains share the same label space with $C$ classes, i.e., $\mathcal{Y}_S = \mathcal{Y}_T$ with $\mathcal{Y}_T$ denoting the target label space. Meanwhile the source and target videos follow different data distributions. Therefore, there exists a domain shift (Ben-David et al., 2006) between $\mathcal{D}_S$ and $\mathcal{D}_T$. The objective of BVDA is to learn a mapping model $\mathcal{V}_T \to \mathcal{Y}_T$ to perform the action recognition task on the unlabeled $\mathcal{D}_T$ while both $\mathcal{D}_S$ and the parameters of $H_S$ are not accessible.

Constrained by the absence of both the source data and the parameters of the source model, neither VUDA methods nor SFVDA methods could be directly applied for BVDA. To tackle BVDA, we resort to an alternative strategy where we adapt target models to the embedded semantic information of the source data by resorting to the hard or soft predictions of the target domain from the black-box source predictor $\hat{\mathcal{Y}}_T = H_S(\mathcal{V}_T)$. Essentially, for BVDA, such a strategy aims to extract effective temporal features with high discriminability and comply with the cluster assumption, where the input distribution contains separated data clusters and that data samples in the same cluster share the same class label (Rigollet, 2007). We therefore propose EXTERN which drives temporal feature towards high discriminability in a self-supervised manner relying on both the *endo-temporal regularization* and the *exo-temporal regularization*. We first introduce the backbone structure of the target model, followed by a thorough illustration over EXTERN and its key components.

## 3.1 Backbone Network

Videos differ from images greatly due to the existence of temporal features. A key prerequisite for the target network to be adapted to the embedded source semantic information is that its backbone could extract temporal features explicitly. An efficient and popular approach constructs the temporal features explicitly with a series of clip features, obtained through clips sampled from the corresponding videos (Wang et al., 2018; Zhou et al., 2018). One notable example is the Temporal Relation Network (TRN) (Zhou et al., 2018). TRN has been widely adopted in previous video domain adaptation tasks, such as VUDA (Chen et al., 2019a; Xu et al., 2022b), PVDA (Xu et al., 2021a) and SFVDA (Xu et al., 2022c) bringing state-of-the-art results. This is thanks to its ability in extracting accurate temporal features by reasoning over correlations between spatial representations which coincides with the process of humans recognizing actions.

Formally, we define an input target video with $k$ frames as $V_i = \{f_i^{(1)}, f_i^{(2)}, ..., f_i^{(k)}\}$, with $f_i^{(j)}$ being the spatial feature of the $j$-th frame in the $i$-th source video. The subscript for the target domain $T$ is omitted for clarity. The spatial features are extracted from the source spatial feature generator $G_{sp}$ which is formulated as a 2D-CNN (e.g., ResNet-50 or ResNet-101 (He et al., 2016; Li et al., 2021)). Subsequently, the temporal feature of $V_i$ is constructed by a combination of multiple clip features, obtained from the temporal feature generator $G_t$. More specifically, $G_t$ builds each clip feature with $r$ temporal-ordered sampled frames where $r \in [2, k]$. Formally, a clip feature for $V_i$, denoted as $cl_i^{(r)}$, is defined by:

$$cl_i^{(r)} = \sum_m g^{(r)}((V_i^{(r)})_m). \tag{1}$$

Here $(V_i^{(r)})_m = \{f_i^{(a)}, f_i^{(b)}, ...\}_m$ is defined as the $m$-th clip with $r$ temporal-ordered frames, where $a$ and $b$ are the frame indices. $a$ and $b$ may not be consecutive as the clip could be extracted with nonconsecutive frames, but should be both constrained within the range of $[1, k]$ with $b > a$. Eventually, the clip feature $cl_i^{(r)}$ is obtained by fusing all the $r$ time ordered frame-level spatial features through an integration function $g^{(r)}$, which is usually formulated as a Multi-Layer Perceptron (MLP). The above computation would result in a total of $(k-1)$ clips. Finally, the temporal feature of $V_i$, denoted as $\mathbf{t}_i$, is computed through the mean aggregation applied across all clip features, defined as:

$$\mathbf{t}_i = \frac{1}{k-1} \sum_r cl_i^{(r)}. \tag{2}$$

## 3.2 Endo and Exo-Temporal Regularizations

**Endo-temporal Regularization** Current methods for BDA (Liang et al., 2022) attempt to obtain features from the unlabeled target data with high discriminability via self-supervised learning by applying

structural regularizations. However, such regularizations are only tailored for spatial features since relevant studies are only conducted over image-based BDA tasks. Comparatively, videos contain temporal features that are constructed from a series of clip features. As depicted in Fig. 1, clip features of discriminative temporal features may still be scattered across the decision boundary thus predicted with different label predictions. This results in indistinct semantic information contained within the temporal features and causes inferior target domain performance. The key to tackling BVDA is to improve the discriminability of clip features by regularizing the behavior of clip features from the same target video. This is achieved by our proposed *endo-temporal regularization* performed over clip features.

More specifically, clip features that are discriminative should meet both the cluster assumption and the *masked-temporal hypothesis*. Notably, the *masked-temporal hypothesis* matches the human intuition that a combination of partial clip features from the same video would still result in consistent prediction as the overall temporal features that combines all clip features if both the temporal features and their clip features are of high discriminability. In other words, the clip features of high discriminability should be clustered toward the corresponding temporal feature.

To attain discriminable clip features, we first augment the target domain with virtual temporal features obtained via a novel yet simple **mask-to-mix** strategy as depicted in Fig. 2. We first define the virtual temporal feature of $V_i$ as $\tilde{\mathbf{t}}_i$. It is constructed by mixing a set of clip features $cl_i^{(r)}$, $r \in [2, k]$ selected by a random masking process. If the temporal feature $\mathbf{t}_i$ is built upon a set of $(k-1)$ clips, there will be exactly $(k-3)$ clips masked randomly, leaving two randomly unmasked clips: $cl_i^{(r1)}$ and $cl_i^{(r2)}$, where $r1, r2 \in [2, k]$. For each mini-batch, the selection of masked clips is random across all input videos within the mini-batch and across each epoch. This is to ensure that the virtual temporal features of $V_i$ are built upon all possible clip pairs from $V_i$ across the whole training process. Different from the temporal feature which is constructed through a simple mean aggregation where all clip features are combined with equal weights, we compute the virtual temporal feature via the MixUp (Zhang et al., 2018) operation. The virtual temporal feature is computed as a linear combination of the two unmasked clips assigned with random weights, defined as:

$$
\begin{aligned}
\tilde{\mathbf{t}}_i &= \text{MixUp}_{\lambda_v}(cl_i^{(r1)}, cl_i^{(r2)}) \\
&= \lambda_v cl_i^{(r1)} + (1 - \lambda_v)cl_i^{(r2)},
\end{aligned}
\tag{3}
$$

where $\lambda_v \in Beta(\alpha_v, \alpha_v)$ is the weight assigned to $cl_i^{(r1)}$ sampled from a Beta distribution with $\alpha_v$ as the hyperparameter. The virtual temporal feature constructed is essentially the linear intermediate representation between the unmasked clip features.

With the constructed virtual temporal feature, the *masked-temporal hypothesis* is satisfied by encouraging the virtual temporal prediction to be consistent with the prediction of the corresponding target video (or equivalently, the corresponding target temporal feature). To achieve such prediction consistency, we aim to minimize the divergence between the virtual temporal prediction and the target prediction. The virtual temporal prediction is obtained from the target predictor $H_T$ directly, i.e., $\tilde{y}_i = H_T(\tilde{\mathbf{t}}_i)$. The target prediction is obtained by applying the target predictor to the target temporal feature, i.e., $y_i = H_T(\mathbf{t}_i)$. The minimization of prediction divergence is formulated as:

$$
\mathcal{L}_{pre} = D_{KL}(\tilde{y}_i \| y_i),
\tag{4}
$$

where $D_{KL}(\cdot \| \cdot)$ denotes the Kullback-Leibler divergence.

To further ensure that the temporal feature contains distinctive semantic information, the clip features should also comply with the cluster assumption, where clip features of the same target video share the same class label. Previous studies (Zhang et al., 2018) suggest that the discriminability of extracted features can be improved and thus the cluster assumption is met if the feature generator behaves linearly in-between training samples. The cluster assumption of clip features is therefore complied by employing the interpolation consistency training (ICT) technique (Verma et al., 2019). Specifically, such a technique encourages the virtual temporal prediction ($\tilde{y}_i$) to be consistent with the mixed unmasked clip prediction. The mixed unmasked clip prediction is computed as the linear combination of the target predictions from both unmasked clips. The mixed unmasked clip prediction is computed as the linear combination of the target predictions

from both unmasked clips, defined as:

$$
\begin{aligned}
y_{mix,i} &= \mathrm{MixUp}_{\lambda_v}(H_T(cl_i^{(r1)}), H_T(cl_i^{(r2)})) \\
&= \lambda_v H_T(cl_i^{(r1)}) + (1 - \lambda_v)H_T(cl_i^{(r2)}).
\end{aligned}
\tag{5}
$$

Subsequently, we aim to optimize the loss function:

$$
\mathcal{L}_{vir} = l_{ce}(\tilde{y}_i, y_{mix,i}),
\tag{6}
$$

where $l_{ce}$ denotes the cross-entropy loss.

Overall, the proposed *endo-temporal regularization* drives the target model to obtain discriminable temporal features by extracting clip features with higher discriminability that comply with the *masked-temporal hypothesis* and the cluster assumption. The *endo-temporal regularization* is applied by jointly optimizing Eq. 4 and Eq. 6:

$$
\mathcal{L}_{endo} = \mathcal{L}_{vir} + \mathcal{L}_{pre}.
\tag{7}
$$

The implementation of the *endo-temporal regularization* can be observed to be very simple, yet it brings significant improvements towards tackling BVDA, as would be presented in Sec. 4.

**Exo-temporal Regularization** To further enhance the discriminability of the temporal feature, we extend the promotion of linear behavior in-between training samples towards the temporal features from different target videos. The temporal features are thus regularized by our proposed *exo-temporal regularization*. Given a pair of videos $V_i, V_j$, we employ the ICT (Verma et al., 2019) across their corresponding temporal features $\mathbf{t}_i, \mathbf{t}_j$. Such operation is equivalent to augmenting the target video domain with interpolated temporal features which would drive the model towards better generalization. Formally, similar to how virtual temporal features are constructed, the interpolated temporal features of $\mathbf{t}_i, \mathbf{t}_j$ are obtained by applying MixUp (Zhang et al., 2018). With $y_i = H_T(\mathbf{t}_i)$ and $y_j = H_T(\mathbf{t}_j)$ denoting the target predictions of $\mathbf{t}_i$ and $\mathbf{t}_j$, the exo-temporal regularization aims to optimize the loss function:

$$
\mathcal{L}_{exo} = l_{ce}(H_T(\mathrm{MixUp}_{\lambda_t}(\mathbf{t}_i, \mathbf{t}_j)), \mathrm{MixUp}_{\lambda_t}(y_i, y_j)),
\tag{8}
$$

where $l_{ce}$ is the cross-entropy loss. $\mathrm{MixUp}_{\lambda_t}$ is defined as shown in Eq. 3 and 5, and $\lambda_t \in Beta(\alpha_t, \alpha_t)$ is the weight assigned to $\mathbf{t}_i$ with $\alpha_t$ as the hyperparameter.

Our mask-to-mix strategy utilizes MixUp for the construction of virtual temporal features, which seeks to obtain consistent prediction as the temporal features such that the corresponding clip features satisfy the *masked-temporal hypothesis*, which is different from existing domain adaptation works based on Mixup that regards it as a data augmentation approach (Xu et al., 2020; Yan et al., 2020; Wu et al., 2020; Panfilov et al., 2019).

### 3.3 Endo and eXo-TEmporal Regularized Network

With the *endo-temporal regularization* and *exo-temporal regularization* terms defined, we propose the EX-TERN to address BVDA leveraging on both regularizations, as depicted in Fig. 2. EXTERN builds upon the TRN backbone structure as specified in Sec. 3.1.

**Extracting Knowledge via Knowledge Distillation.** To extract knowledge from the black-box predictor $H_S$, knowledge distillation (KD) (Hinton et al., 2015) has proven to be an effective solution. The target model is seen as the student, and is trained to learn predictions analogous to that produced by the source model, which is seen as the teacher. However, due to domain shift between the source domain $\mathcal{D}_S$ and target domain $\mathcal{D}_T$, the output from the source model could be noisy and inaccurate. To cope with such drawback, the **adaptive label smoothing** (AdaLS) technique (Liang et al., 2022) with self-distillation (Laine & Aila, 2017) and exponential moving average (EMA) (Grebenkov & Serror, 2014) update is recently proposed. Here, only the predictions of the top-$c$ classes are directly utilized while predictions of other classes are

forced to a uniform distribution as:

$$\hat{y}_i' = \text{AdaLS}_c(\hat{y}_i) = \begin{cases} \hat{y}_i^p, & p \in \mathcal{T}_i^c \\ \frac{1 - \sum_{q \in \mathcal{T}_i^c} \hat{y}_i^q}{C - c}, & \text{otherwise,} \end{cases} \tag{9}$$

where $\hat{y}_i \in \hat{\mathcal{Y}}$ is the prediction of the target video $V_i$ obtained from the black-box source predictor $H_S$ (i.e., the teacher prediction), while $y_i^p$ denotes the prediction of the $p$-th class and $\mathcal{T}_i^c$ denotes the class index set of the top-$c$ predictions for input $V_i$. The teacher prediction is further dynamically updated per training epoch to maintain a EMA prediction. We apply AdaLS with EMA to reduce noisy information by focusing only on the top-predicted classes. Extracting source knowledge is eventually achieved by optimizing:

$$\mathcal{L}_{kd} = \mathbb{E}_{V_i \in \mathbf{D}_T} D_{KL}(\hat{y}_i' \| y_i). \tag{10}$$

**Learning Adaptive Clip Weights.** Previous research (Nguyen et al., 2020; Chen et al., 2019a) shows that features with lower prediction uncertainty would possess higher discriminability. Therefore, to better aggregate the clip features for the temporal feature, we assign a **clip weight** to each clip feature by attending to clip features with lower prediction uncertainty. Specifically, the clip weight is defined as the additive inverse of the target predictions of the corresponding clip, computed as:

$$w_{cl_i^{(r)}} = 1 - h(H_T(cl_i^{(r)})), \tag{11}$$

where the constant 1 is added as a residual connection for more stable optimization. Subsequently, the temporal feature is obtained as the weighted aggregation of all clip features, with Eq. 2 modified as:

$$\mathbf{t}_i = \frac{1}{k-1} \sum_r w_{cl_i^{(r)}} \, cl_i^{(r)}. \tag{12}$$

**Information Maximization.** Inspired by prior works in BDA (Liang et al., 2022; Yang et al., 2022), we maximize the **mutual information** (MI) to encourage diversity among target predictions and to promote their individual certainty:

$$\mathcal{L}_{mi} = h(\mathbb{E}_{V_i \in \mathbf{D}_T} y_i) - \mathbb{E}_{V_i \in \mathbf{D}_T} h(y_i), \tag{13}$$

where $y_i = H_T(G_{sp}(G_t(V_i))) = H_T(\mathbf{t}_i)$ is the target prediction for input $V_i$ and $h(y_i) = -\sum_{c=1}^{C} y_i^c \log y_i^c$ is the conditional entropy function. Maximizing MI could **marginally** improve the performances for BVDA, as would be shown later in the ablation studies (Sec. 4.3).

The aforementioned techniques of leveraging AdaLS, adaptive clip weights and information maximization have been proven to be effective for black-box domain adaptation for images (Liang et al., 2022) and source-free video unsupervised domain adaptation (Xu et al., 2022c). They are leveraged to build a strong baseline for BVDA.

Table 1: Results for BVDA on UCF-HMDB$_{full}$ and Sports-DA for closed-set video domain adaptation.

| Methods | Publication | Privacy | | UCF-HMDB$_{full}$ | | | Sports-DA | | | | | | |
|---|---|---|---|---|---|---|---|---|---|---|---|---|---|
| | | Data | Model | U101→H51 | H51→U101 | Avg. | K600→U101 | K600→S1M | S1M→U101 | S1M→K600 | U101→K600 | U101→S1M | Avg. |
| TRN (Zhou et al., 2018) | ECCV-18 | - | - | 76.11 | 78.97 | 77.54 | 90.25 | 71.16 | 88.95 | 73.90 | 62.73 | 49.74 | 72.79 |
| LNL (Zhang et al., 2021) | - | ✓ | ✓ | 75.78 | 78.92 | 77.35 | 82.37 | 68.44 | 82.11 | 73.11 | 59.03 | 54.84 | 69.98 |
| HD-SHOT (Liang et al., 2021) | TPAMI(21') | ✓ | ✓ | 77.86 | 80.39 | 79.13 | 87.08 | 69.75 | 81.59 | 72.11 | 65.63 | 60.49 | 72.78 |
| SD-SHOT (Liang et al., 2021) | TPAMI(21') | ✓ | ✓ | 79.29 | 82.22 | 80.76 | 85.39 | 68.07 | 83.58 | 74.80 | 63.94 | 60.75 | 72.75 |
| DINE (Liang et al., 2022) | CVPR-21 | ✓ | ✓ | 81.39 | 87.57 | 84.48 | 91.60 | 72.11 | 86.54 | 77.59 | 76.22 | 66.95 | 78.50 |
| EXTERN | - | ✓ | ✓ | **88.89** | **91.95** | **90.42** | **93.77** | **73.79** | **95.42** | **82.16** | **81.19** | **72.74** | **83.18** |
| TA³N (Chen et al., 2019a) | ICCV-19 | ✗ | ✗ | 77.70 | 85.37 | 81.54 | 90.28 | 68.57 | 92.97 | 72.65 | 63.63 | 54.06 | 73.70 |
| DANN (Ganin & Lempitsky, 2015) | ICML-15 | ✗ | ✗ | 78.63 | 90.29 | 84.46 | 87.97 | 75.05 | 85.75 | 73.40 | 65.88 | 55.08 | 73.85 |
| MK-MMD (Long et al., 2015) | ICML-15 | ✗ | ✗ | 77.99 | 86.18 | 82.09 | 90.16 | 67.95 | 90.95 | 73.58 | 66.10 | 55.58 | 74.05 |
| SAVA (Choi et al., 2020) | ECCV-20 | ✗ | ✗ | 78.56 | 89.28 | 83.92 | 97.33 | 75.76 | 91.20 | 75.28 | 58.17 | 51.33 | 74.85 |
| SHOT (Liang et al., 2021) | TPAMI(21') | ✓ | ✗ | 77.44 | 86.77 | 82.10 | 91.91 | 72.44 | 91.95 | 75.57 | 67.81 | 52.11 | 75.30 |
| ACAN (Xu et al., 2022b) | - | ✗ | ✗ | 84.04 | 93.78 | 88.91 | 94.70 | 76.69 | 92.32 | 77.69 | 62.50 | 52.38 | 76.05 |
| ATCoN (Xu et al., 2022c) | ECCV-22 | ✓ | ✗ | 83.21 | 91.07 | 87.14 | 97.59 | 77.56 | 94.36 | 80.32 | 67.20 | 55.17 | 78.70 |

**Overall Objective.** Summarizing all the loss functions as presented in Eqs. (7, 8, 10, 13), the overall optimization objective of EXTERN is expressed as:

$$\mathcal{L} = \mathcal{L}_{kd} + \beta_{reg}(\mathcal{L}_{endo} + \mathcal{L}_{exo}) - \mathcal{L}_{mi}, \tag{14}$$

where $\beta_{reg}$ is the hyperparameter that controls the strength of the regularizations and is empirically set to 1. We refer to the settings of DINE (Liang et al., 2022) by setting $\alpha_t$ as 0.3 and $c$ as 3.

Table 2: Results for BVDA on Daily-DA for closed-set video domain adaptation.

| Methods | Privacy | | Daily-DA | | | | | | | | | | | | |
|---|---|---|---|---|---|---|---|---|---|---|---|---|---|---|---|
| | Data | Model | K600→A11 | K600→H51 | K600→MIT | MIT→A11 | MIT→H51 | MIT→K600 | H51→A11 | H51→MIT | H51→K600 | A11→H51 | A11→MIT | A11→K600 | Avg. |
| TRN (Zhou et al., 2018) | - | - | 25.91 | 37.50 | 31.25 | 20.25 | 45.83 | 61.66 | 16.99 | 33.25 | 43.45 | 20.42 | 13.25 | 21.66 | 30.95 |
| LNL (Zhang et al., 2021) | ✓ | ✓ | 20.75 | 49.38 | 32.25 | 15.51 | 41.52 | 55.96 | 16.80 | 31.75 | 41.34 | 20.04 | 14.00 | 35.85 | 31.26 |
| HD-SHOT (Liang et al., 2021) | ✓ | ✓ | 15.84 | 46.87 | 32.50 | 16.26 | 39.14 | 56.52 | 15.87 | 31.00 | 43.12 | 23.28 | 15.25 | 42.60 | 31.52 |
| SD-SHOT (Liang et al., 2021) | ✓ | ✓ | 17.02 | 47.92 | 33.25 | 16.56 | 41.07 | 58.16 | 16.17 | 32.50 | 46.96 | 24.49 | 16.00 | 45.57 | 32.97 |
| DINE (Liang et al., 2022) | ✓ | ✓ | 19.47 | 50.83 | 34.50 | 14.28 | 49.17 | 64.00 | 23.51 | 38.75 | 51.17 | 21.25 | 17.75 | 47.03 | 35.98 |
| EXTERN | ✓ | ✓ | **23.97** | **55.83** | **35.25** | **18.15** | **53.75** | **68.14** | **26.22** | **40.75** | **57.66** | **26.25** | **18.25** | **51.45** | **39.64** |
| TA³N (Chen et al., 2019a) | ✗ | ✗ | 23.51 | 36.17 | 31.75 | 18.94 | 43.77 | 57.19 | 16.58 | 28.75 | 40.38 | 17.81 | 14.00 | 22.04 | 29.24 |
| DANN (Ganin & Lempitsky, 2015) | ✗ | ✗ | 25.30 | 38.34 | 23.25 | 20.71 | 45.30 | 61.86 | 16.86 | 35.25 | 40.26 | 24.46 | 19.00 | 27.38 | 31.50 |
| MK-MMD (Long et al., 2015) | ✗ | ✗ | 25.88 | 37.06 | 25.75 | 19.09 | 52.71 | 61.57 | 24.16 | 30.75 | 35.58 | 22.81 | 17.25 | 26.40 | 31.58 |
| SAVA (Choi et al., 2020) | ✗ | ✗ | 26.33 | 38.29 | 32.00 | 20.61 | 46.50 | 62.64 | 21.30 | 34.00 | 44.38 | 23.74 | 13.50 | 22.08 | 32.11 |
| SHOT (Liang et al., 2021) | ✓ | ✗ | 18.37 | 48.40 | 34.50 | 13.88 | 38.33 | 53.73 | 22.05 | 29.00 | 47.92 | 31.93 | 16.50 | 39.52 | 32.84 |
| ACAN (Xu et al., 2022b) | ✗ | ✗ | 27.08 | 42.39 | 33.50 | 21.17 | 47.97 | 63.88 | 21.81 | 34.75 | 45.79 | 25.35 | 15.00 | 31.73 | 34.20 |
| ATCoN (Xu et al., 2022c) | ✓ | ✗ | 22.55 | 53.32 | 35.00 | 24.73 | 52.50 | 65.90 | 25.28 | 36.75 | 53.51 | 32.44 | 17.00 | 43.45 | 38.54 |

# 4 Experiments

In this section, we evaluate our proposed EXTERN across a variety of cross-domain action recognition benchmarks, covering a wide range of cross-domain scenarios. We demonstrate exceptional performances on all benchmarks. Moreover, thorough ablation studies and analysis of EXTERN are performed to further justify the design of EXTERN.

## 4.1 Experimental Settings

**Datasets.** We evaluate EXTERN on three benchmarks: UCF-HMDB$_{full}$ (Chen et al., 2019a), Sports-DA (Xu et al., 2023) and Daily-DA (Xu et al., 2023). **UCF-HMDB$_{full}$** is one of the most common benchmarks for VUDA and is constructed from UCF101 (U101) (Soomro et al., 2012) and HMDB51 (H51) (Kuehne et al., 2011), each corresponding to a separate domain. **Sports-DA** is a large-scale benchmark with three domains, built from UCF101, Sports-1M (S1M) (Karpathy et al., 2014), and Kinetics (K600). **Daily-DA** incorporates both normal and low-illumination videos with four domains, built from ARID (A11) (Xu et al., 2021b) (a low-illumination video dataset), HMDB51, Moments-in-Time (MIT) (Monfort et al., 2019), and Kinetics (Kay et al., 2017). The distant domain shift due to immense illumination difference renders it more challenging.

**Implementation.** We implement our method with the PyTorch (Paszke et al., 2019) library. To obtain video features, we instantiate Temporal Relation Network (TRN) (Zhou et al., 2018) with ResNet-50 (He et al., 2016) as the model backbone for both the black-box source model and the target model. The TRN is leveraged thanks to its capability in extracting explicit temporal features via reasoning over correlations between spatial representations which coincides with how humans recognize actions. The TRN has therefore been widely adopted in previous video unsupervised domain adaptation tasks, including closed-set video domain adaptation (Chen et al., 2019a; Xu et al., 2022b), multi-set video domain adaptation (Xu et al., 2023), and partial-set video domain adaptation (Xu et al., 2021a), delivering state-of-the-art results in the respective tasks. For the source model, an additional fully connected layer is inserted before the last fully connected layer which acts as the classifier. For the target model, following DINE (Liang et al., 2022), a Batch Normalization (Ioffe & Szegedy, 2015) and an additional fully connected layer are inserted before the final classifier, which is constructed with a weight normalization (Salimans & Kingma, 2016) layer and a fully connected layer.

Similar strategies are applied for training the black-box source model and the target model, where the TRN backbones are both initialized from pre-trained weights obtained by pre-training on ImageNet (Deng et al., 2009). All new layers are trained from scratch, with their learning rates set to be 10 times that of the pretrained-loaded layers. For the black-box source model, the training lasts for 100 epochs for tasks related to the Sports-DA and the MiniKinetics-UCF dataset, and for 50 epochs for all other datasets. For the target model, the training lasts for 20 epochs for tasks related to the UCF-HMDB$_{full}$ dataset and the UCF-HMDB$_{partial}$ dataset, 30 epochs for tasks related to the Daily-DA dataset and the HMDB-ARID$_{partial}$ dataset, and 50 epochs for the Sports-DA dataset and the MiniKinetics-UCF dataset. The stochastic gradient descent (SGD) algorithm (Bottou, 2010) is used for optimization, with the weight decay set to 0.0001 and the momentum set to 0.9. The batch size is set to 32 input videos per GPU. Hyper-parameters $\alpha_v = 0.3$ and $\beta_{reg} = 1.0$ are empirically set and fixed.

**Baselines.** We compare EXTERN with state-of-the-art BDA approaches as well as competitive UDA/VDA approaches. For fair comparisons, all compared methods are re-implemented with the exact same backbone as EXTERN. Specifically, for BDA approaches, **LNL** (Zhang et al., 2021) is a noisy label learning method where pseudo labels are refined with KL divergence and leveraged for iterative network training. **HD-SHOT** and **SD-SHOT** obtain the model through self-training and apply SHOT (Liang et al., 2021) by employing a cross-entropy loss and weighted cross-entropy loss respectively. We also compare with methods including: DINE (Liang et al., 2022), DANN (Ganin & Lempitsky, 2015), MK-MMD (Long et al., 2015), TA$^3$N (Chen et al., 2019a), SAVA (Choi et al., 2020), ACAN (Xu et al., 2022b), SHOT (Liang et al., 2021), ATCoN (Xu et al., 2022c), BA$^3$US (Liang et al., 2020b), PADA (Cao et al., 2018) and PATAN (Xu et al., 2021a). We report the average accuracies over five runs with identical settings.

Table 3: Results for BVDA on UCF-HMDB$_{partial}$, HMDB-ARID$_{partial}$ and MiniKinetics-UCF for partial-set video domain adaptation.

| Methods | Publication | Privacy | | UCF-HMDB$_{partial}$ | | | HMDB-ARID$_{partial}$ | | | MiniKinetics-UCF | | |
|---|---|---|---|---|---|---|---|---|---|---|---|---|
| | | Data | Model | U-14→H-7 | H-14→U-7 | Avg. | H-10→A-5 | A-10→H-5 | Avg. | M-45→U-18 | U-45→M-18 | Avg. |
| TRN (Zhou et al., 2018) | ECCV-18 | - | - | 59.05 | 82.33 | 70.69 | 21.54 | 29.33 | 25.44 | 64.30 | 87.56 | 75.93 |
| LNL (Zhang et al., 2021) | - | ✓ | ✓ | 56.79 | 80.94 | 68.86 | 22.23 | 26.57 | 24.40 | 61.40 | 85.92 | 73.66 |
| HD-SHOT (Liang et al., 2021) | TPAMI(21') | ✓ | ✓ | 56.41 | 80.62 | 68.51 | 23.30 | 26.84 | 25.07 | 59.95 | 89.62 | 74.78 |
| SD-SHOT (Liang et al., 2021) | TPAMI(21') | ✓ | ✓ | 61.52 | 82.42 | 71.97 | **23.74** | 25.62 | 24.68 | 61.07 | 88.79 | 74.93 |
| DINE (Liang et al., 2022) | CVPR-21 | ✓ | ✓ | 66.19 | 83.84 | 75.01 | 17.69 | 17.33 | 17.51 | 68.79 | 93.56 | 81.18 |
| EXTERN | - | ✓ | ✓ | **71.43** | **90.60** | **81.02** | 23.08 | **38.67** | **30.87** | **75.89** | **96.49** | **86.19** |
| TA$^3$N (Chen et al., 2019a) | ICCV-19 | ✗ | ✗ | 50.99 | 73.70 | 62.35 | 20.95 | 27.08 | 24.02 | 63.24 | 92.14 | 77.69 |
| DANN (Ganin & Lempitsky, 2015) | ICML-15 | ✗ | ✗ | 61.56 | 77.63 | 69.59 | 22.73 | 19.54 | 21.13 | 62.06 | 93.04 | 77.55 |
| MK-MMD (Long et al., 2015) | ICML-15 | ✗ | ✗ | 59.16 | 82.25 | 70.70 | 22.31 | 25.79 | 24.05 | 69.26 | 88.69 | 78.98 |
| SAVA (Choi et al., 2020) | ECCV-20 | ✗ | ✗ | 54.74 | 83.41 | 69.08 | 25.27 | 27.94 | 26.61 | 66.49 | 90.31 | 78.40 |
| PADA (Cao et al., 2018) | ECCV-18 | ✗ | ✗ | 68.37 | 85.86 | 77.11 | 21.28 | 32.60 | 26.94 | 72.72 | 91.62 | 82.17 |
| BA$^3$US (Liang et al., 2020b) | ECCV-20 | ✗ | ✗ | 71.85 | 88.41 | 80.13 | 26.81 | 32.20 | 29.51 | 76.41 | 95.44 | 85.93 |
| PATAN (Xu et al., 2021a) | ICCV-21 | ✗ | ✗ | 73.60 | 91.85 | 82.72 | 30.34 | 35.51 | 32.93 | 77.31 | 96.50 | 86.90 |

## 4.2 Overall Results and Comparisons

**Closed-set Domain Adaptation.** We show the results on UCF-HMDB$_{full}$ and Sports-DA in Tab. 1, and results on Daily-DA in Tab. 2. Our proposed EXTERN achieve state-of-the-art results across all the three cross-domain benchmarks. On average, EXTERN outperforms all BDA approaches designed for image-based DA tasks (i.e., LNL, HD/SD-SHOT and DINE), outperforming the best method by a relative 7.0%, 6.0% and 10.2% respectively. This justifies the effectiveness of the designed regularizations tailored for temporal features whose discriminability relies on clip features complying with the *masked-temporal hypothesis* and the cluster assumption. It could also be observed that the prior BDA approaches may fail to tackle BVDA well, with at least one task of Sports-DA and Daily-DA benchmarks showing inferior performance to that of the source-only model. Prior BDA approaches focused solely on spatial features, and may not obtain clip features that meet the *masked-temporal hypothesis* and temporal feature with distinct semantic information. This result in negative impacts compared to the source-only baseline. Notably, EXTERN even outperforms various VUDA approaches (Chen et al., 2019a; Choi et al., 2020; Xu et al., 2022b) with source data accessibility. Since EXTERN performs adaptation based solely on prediction results, it is found that EXTERN would not be affected by noise contained within the source data or the source model, generating superior adaptation results. This shows that training a target model from scratch with strong regularizations while adapting solely with source predictions can be as effective as data-based domain alignment techniques.

**Partial-set Domain Adaptation.** Apart from closed-set video domain adaptation, we further demonstrate the generalization ability of our proposed EXTERN by evaluating on partial-set video domain adaptation (PVDA) tasks. To achieve this, we follow (Xu et al., 2021a) and leverage on three other benchmarks: UCF-HMDB$_{partial}$, HMDB-ARID$_{partial}$ and MiniKinetics-UCF. Specifically, **UCF-HMDB$_{partial}$** is built from UCF101 and HMDB51 from 14 overlapping categories and contains two PVDA tasks: **U-14→H-7** and **H-14→U-7**. **HMDB-ARID$_{partial}$** is built from HMDB51 and ARID, which is more challenging thanks to the distant domain shift. The dataset is collected from 10 and contains two PVDA tasks: **H-10→A-5** and **A-10→H-5**. **MiniKinetics-UCF** is a large-scale dataset built from MiniKinetics (Xie et al., 2017) and UCF101 containing 45 overlapping categories, also containing two PVDA tasks: **M-45→U-18** and **U-45→M-18**.

Table 4: Ablation studies of learning objectives and clip weights on UCF-HMDB$_{full}$ and UCF-HMDB$_{partial}$.

| Methods | Components | | | | | | UCF-HMDB$_{full}$ | | UCF-HMDB$_{partial}$ | | Avg. |
|---|---|---|---|---|---|---|---|---|---|---|---|
| | $\mathcal{L}_{kd}$ | $\mathcal{L}_{mi}$ | $\mathcal{L}_{exo}$ | $\mathcal{L}_{vir}/\mathcal{L}_{endo}$ | $\mathcal{L}_{pre}/\mathcal{L}_{endo}$ | $w_{cl}$ | U101→H51 | H51→U101 | U-14→H-7 | H-14→U-7 | |
| EXTERN | ✓ | ✓ | ✓ | | | | 78.25 | 85.64 | 60.72 | 81.09 | 76.42 |
| | ✓ | ✓ | ✓ | | | ✓ | 80.08 | 86.34 | 61.91 | 82.71 | 77.76 |
| | ✓ | ✓ | ✓ | | ✓ | | 82.66 | 89.49 | 64.52 | 86.47 | 80.78 |
| | ✓ | ✓ | ✓ | | ✓ | ✓ | 83.89 | 89.93 | 65.24 | 88.16 | 81.80 |
| | ✓ | ✓ | ✓ | ✓ | | | 86.47 | 90.63 | 68.11 | 88.23 | 83.36 |
| | ✓ | ✓ | ✓ | ✓ | | ✓ | 87.22 | 90.89 | 69.05 | 89.47 | 84.16 |
| | ✓ | ✓ | | ✓ | ✓ | | 84.65 | 90.11 | 66.62 | 87.59 | 82.24 |
| | ✓ | ✓ | | ✓ | ✓ | ✓ | 85.83 | 90.37 | 67.71 | 88.91 | 83.21 |
| | | ✓ | ✓ | ✓ | ✓ | ✓ | 87.74 | 91.16 | 70.19 | 90.08 | 84.79 |
| | ✓ | | ✓ | ✓ | ✓ | ✓ | 88.43 | 91.64 | 71.06 | 90.34 | 85.36 |
| | | | ✓ | ✓ | ✓ | ✓ | 87.65 | 91.16 | 70.04 | 89.84 | 84.67 |
| | ✓ | ✓ | ✓ | ✓ | ✓ | | 87.92 | 91.33 | 70.38 | 90.08 | 84.93 |
| | ✓ | ✓ | ✓ | ✓ | ✓ | ✓ | **88.89** | **91.95** | **71.43** | **90.60** | **85.72** |

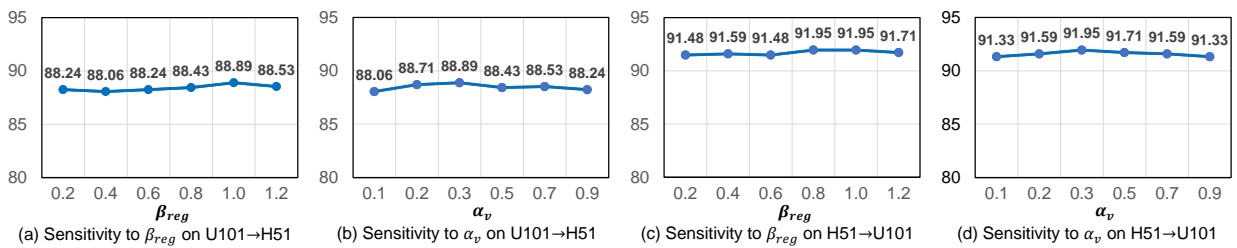

Figure 3: Sensitivity of hyperparameters $\beta_{reg}$ and $\alpha_v$ on UCF-HMDB$_{full}$.

(a) Sensitivity to $\beta_{reg}$ on U101→H51
(b) Sensitivity to $\alpha_v$ on U101→H51
(c) Sensitivity to $\beta_{reg}$ on H51→U101
(d) Sensitivity to $\alpha_v$ on H51→U101

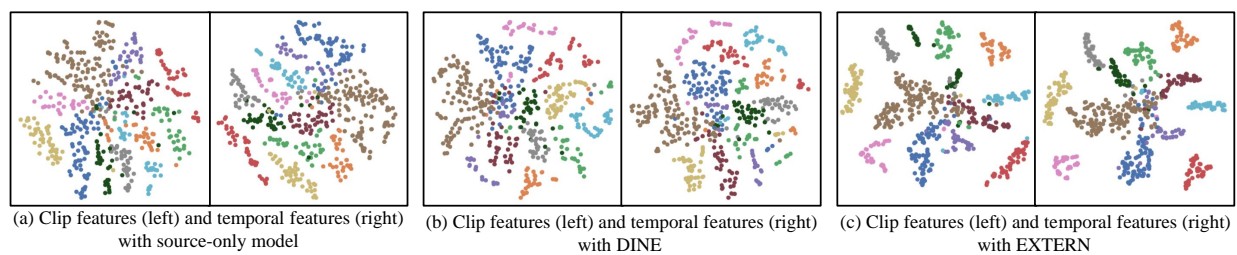

(a) Clip features (left) and temporal features (right) with source-only model
(b) Clip features (left) and temporal features (right) with DINE
(c) Clip features (left) and temporal features (right) with EXTERN

Figure 4: t-SNE (Van der Maaten & Hinton, 2008) visualizations of clip features and temporal features extracted by (a) source-only model, (b) DINE (Liang et al., 2022), and (c) EXTERN, with class information. Different colors denotes different classes.

The results for partial-set cross-domain action recognition are presented in Tab. 3. Partial-set is more challenging due to the asymmetric label spaces with the existence of "source-only" classes, causing negative transfer (Cao et al., 2018; Xu et al., 2021a). Negative transfer affects all previous BDA approaches, where all approaches will under-perform against the source-only baseline in at least one benchmark. Despite such challenge, EXTERN still achieves outstanding results, outperforming the best BDA approach by a relative 8.0%, 23.1% and 6.2% on the three benchmarks respectively. EXTERN also surpasses several PDA approaches even though EXTERN is not specifically catered for label shift.

## 4.3 Ablation Studies and Analysis

To gain a deeper understanding of the effectiveness of EXTERN while justifying its design, we perform detailed ablation studies as shown in Tab. 4 and Fig. 3. Specifically, the ablations studies explore EXTERN from two perspectives: the effects of individual learning objectives and the effects of assigning clip weights.

Table 5: Ablation studies of regularization weights on UCF-HMDB$_{full}$ and UCF-HMDB$_{partial}$.

| Methods | Weights | | UCF-HMDB$_{full}$ | | UCF-HMDB$_{partial}$ | | Avg. |
|---|---|---|---|---|---|---|---|
| | $\beta_{reg}^{ex}$ | $\beta_{reg}^{en}$ | U101→H51 | H51→U101 | U-14→H-7 | H-14→U-7 | |
| | 1.0 | 1.0 | 88.89 | 91.95 | 71.43 | 90.60 | **85.72** |
| | 1.0 | 0.0 | 80.08 | 86.34 | 61.91 | 82.71 | 77.76 |
| | 1.0 | 0.1 | 82.75 | 88.49 | 64.57 | 84.34 | 80.04 |
| | 1.0 | 0.5 | 86.35 | 91.59 | 69.02 | 88.20 | 83.79 |
| EXTERN | 1.0 | 1.5 | 88.53 | **92.12** | **72.48** | 89.42 | 85.64 |
| | 1.0 | 2.0 | **89.28** | 91.42 | 70.80 | **90.84** | 85.59 |
| | 0.0 | 1.0 | 85.83 | 90.37 | 67.71 | 88.91 | 83.21 |
| | 0.1 | 1.0 | 86.44 | 90.84 | 68.45 | 89.33 | 83.76 |
| | 0.5 | 1.0 | 88.43 | 91.48 | 70.98 | 90.31 | 85.30 |
| | 1.5 | 1.0 | 88.24 | 90.84 | 71.75 | 90.18 | 85.25 |
| | 2.0 | 1.0 | 87.92 | 91.10 | 70.42 | 89.88 | 84.83 |

The ablation studies are conducted on UCF-HMDB$_{full}$ and UCF-HMDB$_{partial}$ with the same TRN backbone as previous experiments.

**Endo and Exo-Temporal Regularizations.** As demonstrated in Tab. 4, there is a notable performance drop when either $\mathcal{L}_{exo}$, $\mathcal{L}_{endo}$ or any of the components of $\mathcal{L}_{endo}$ (i.e., $\mathcal{L}_{vir}$ and $\mathcal{L}_{pre}$) is removed from the learning objective, thus justifying that the designed learning objectives complement each other. Further, by applying the proposed endo and exo-temporal regularizations alone (optimizing $\mathcal{L}_{endo}$, $\mathcal{L}_{vir}$, and $\mathcal{L}_{pre}$), EXTERN could outperform all prior BDA approaches and even some UDA/PDA approaches. This further proves the effectiveness of both the *endo-temporal regularization* and *exo-temporal regularization* since these regularizations are tailored to temporal features.

Meanwhile, the weights of both the endo- and exo-temporal regularizations are first set equally as $\beta_{reg}$. A larger performance drop occurs when the endo-temporal regularization is not applied. Therefore, it is intuitive that the two regularization terms may have distinct contributions to the final performance. A detailed comparison over the contribution of the different regularizations is presented in Tab. 5, where the weights of the exo- and endo-temporal regularizations are denoted as $\beta_{reg}^{ex}$ and $\beta_{reg}^{en}$ respectively. The results suggests that the endo-temporal regularization would contribute more towards EXTERN's performance. This is thanks to its capability in pushing clip features towards the *masked-temporal hypothesis*. When $\beta_{reg}^{ex} = 1.0$ and $\beta_{reg}^{en} \leqslant 1.0$, a small weight increase in the endo-temporal regularization would result in notable performance improvement of EXTERN. EXTERN achieves the overall best performance when both regularizations are balanced, further increasing $\beta_{reg}^{en}$ would achieve better results in certain adaptation tasks.

**Knowledge Distillation, Information Maximization, and Clip Weight.** Tab. 4 also shows that applying knowledge distillation and information maximization brings further improvements towards EXTERN. However, the scale of which brought by these techniques is marginal compared to applying the temporal feature-tailored regularizations. Meanwhile, results also justify the need to construct temporal features attentively with clip weight, bringing consistent performance gain, though the gain is also relatively marginal.

**Hyperparameter Sensitivity.** We focus on studying the hyperparameter sensitivity of $\beta_{reg}$ which controls the strength of the regularizations and $\alpha_v$ which relates to the construction of virtual temporal feature $\tilde{\mathbf{t}}_i$. Here $\beta_{reg}$ is in the range of 0.2 to 1.2 and $\alpha_v$ is in the range of 0.1 to 0.9. As shown in Fig. 3, the results of EXTERN falls within a marginal 0.83% which ranges from 88.06% to 88.89% for the U101→H51 task, and a marginal 0.62% which ranges from 91.33% to 91.95% for the H51→U101 task. EXTERN obtains the best results for both tasks at $\alpha_v = 0.3$ and $\beta_{reg} = 1.0$. The minimal variations show that the performance of EXTERN is robust to both hyperparameters. Meanwhile, despite the slight variations, EXTERN maintains the best results with all the hyperparameter settings.

**Feature Visualization.** We further understand the characteristics of EXTERN by plotting the t-SNE embeddings (Van der Maaten & Hinton, 2008) of both the clip features and temporal features extracted by the source-only model, DINE and EXTERN for H51→U101, as shown in Fig. 4. It is clearly observed that both the clip features and temporal features from EXTERN are more clustered and discriminable, justifying

Table 6: Detailed comparison of EXTERN with related but different VUDA and UDA methods.

| Method | Publication | Task | Techniques |
|---|---|---|---|
| DINE (Liang et al., 2022) | CVPR-21 | Black-box Domain Adaptation (BDA): source image data not available, source image model provided as a black-box predictor whose parameters are not available, target label not available, image-based. | DINE extracts target image features by obtaining pseudo-labels from the black-box predictors while applying structural regularizations. |
| ATCoN (Xu et al., 2022c) | ECCV-22 | Source-free Video Domain Adaptation (SFVDA): source video data not available, source video model parameters are available, target label not available, video-based. | (a) ATCoN is designed such that the source and target models are identical in their structure, where the target model leverages the source classifier directly; (b) ATCoN learns temporal consistency which includes feature consistency and source prediction consistency across local temporal features; (c) ATCoN attends to local temporal features based on prediction confidence obtained from source classifier. |
| TA$^3$N (Chen et al., 2019a) | ICCV-19 | Video Unsupervised Domain Adaptation (VUDA): source video data and source video model are available, target label not available, video-based. | (a) TA$^3$N (Chen et al., 2019a) aligns source and target videos by applying adversarial-based domain adaptation with domain discriminators across both spatial and local temporal features; (b) TA$^3$N (Chen et al., 2019a) attends to the local temporal features with high domain discriminability. |
| DM-ADA (Xu et al., 2020) | AAAI-20 | Unsupervised Domain Adaptation (UDA): source image data and source image model are available, target label not available, image-based. | (a) DM-ADA leverages domain MixUp which augments the target domain with source domain data. (b) DM-ADA utilizes soft domain labels to improve the generalization ability of the feature extractor and obtain a domain discriminator judging samples' difference relative to two domains with refined scores. |
| **EXTERN (Ours)** | - | Black-box Video Domain Adaptation (BVDA): source video data **not** available, source video model provided as a black-box predictor whose parameters are **not** available, target label not available, video-based | (a) EXTERN extracts temporal features in a self-supervised manner by applying the *endo-temporal regularization* and the *exo-temporal regularization*; (b) EXTERN distills knowledge from the predictions obtained from the source predictor; (c) The endo-temporal regularization drives clip features towards satisfying the cluster assumption (Rigollet, 2007) and the *masked-temporal hypothesis* by augmenting the target video domain with virtual temporal features through a **mask-to-mix** strategy over clip features. |

that the applied regularizations can promote higher discriminability and better compliance with the cluster assumption. We can also observe that the distribution of clip features is more similar to the distribution of temporal features with EXTERN. This intuitively proves that EXTERN drives clip features towards satisfying the *masked-temporal hypothesis* where clip features are aligned towards the temporal features, and ensures that the temporal features contain distinct semantic information with high discriminability.

## 4.4 Detail Comparison with Related VUDA and UDA Methods

To highlight the novelty of EXTERN, we further compare our EXTERN with previous VUDA and UDA methods in detail. Specifically, we compare with DINE (Liang et al., 2022), ATCoN (Xu et al., 2022c), TA$^3$N (Chen et al., 2019a), and DM-ADA (Xu et al., 2020) which is an image-based UDA method that leverages MixUp (Zhang et al., 2018). The methods are compared from two perspectives: the tasks they tackle and the techniques leveraged, as shown in Table 6.

## 5   Conclusion and Future Work

In this work, we pioneer in formulating and exploring the realistic yet more challenging task of *Black-box Video Domain Adaptation* (BVDA) for privacy-preserving and portable video model transfer. We propose EXTERN for BVDA which obtains effective and discriminative temporal features by driving clip features to satisfy the *masked-temporal hypothesis* and the cluster assumption. This is achieved by applying a novel *endo-temporal regularization* following a mask-to-mix strategy, along with an exo-temporal regularization. Results across multiple cross-domain action recognition benchmarks under both closed-set and partial-set domain adaptation settings justify the efficacy of EXTERN. We believe that such a superior performance of EXTERN could pave a new way for tackling video domain adaptation without compromising data privacy.

While the proposed EXTERN has proven to be effective for the BVDA task, there is still room for future improvements. We observe that EXTERN may not perform well in tasks where the domain gap is relatively large, as depicted in Table 2 for tasks involving the ARID domain. This suggest that the current endo- and exo-temporal regularizations which are built on features constructed with the linear MixUp approach may not work under large domain shift scenarios. Virtual temporal features and interpolated temporal features constructed via higher order computation could be explored. Further, the temporal relationship between clip features could also be leveraged more explicitly as an additional modality for adaptation. Additionally, current BVDA tasks have only been explored for the closed-set and partial-set scenarios. More challenging settings such as the open-set setting where the target video domain may contain target-private classes may also be further explored.

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
