# OpenReview forum: "Leveraging Endo- and Exo-Temporal Regularization for Black-box Video Domain Adaptation"
_TMLR — Accepted by TMLR_

### Review · Reviewer_pFxC · 2023-11-01

**Summary Of Contributions:**

This paper concerns black box domain adaptation for video where source data are not available and the model is provided only as a black box predictor (no parameters/weights available). Also the target labels are not available. \

What this paper proposes is to create two regularisation methods termed endotemporal and exotemporal regularisation. \

The contributions are:
a) The two new regularisation terms that are used to extract temporal features in a self-supervised manner \
b) The method is using a distillation approach to extract knowledge from the predictions obtained from the source predictor alone \
c) results and experiments are extensive and show very good performance across several benchmark datasets and ablations

**Audience:**

Yes

**Claims And Evidence:**

Yes

**Requested Changes:**

a) Clarify needs to be improved - the whole paper seems to be contextualised around the two regularisation terms so this needs to be fleshed out more clearly \
b) figure 2 needs fixing; maybe another figure can be added to show how the two regularisation terms work \
c) there are several typos and long sentences. \
d) Conclusion is short; please expand and provide a short discussion on importance, generalisation and future perspectives

**Strengths And Weaknesses:**

Pros
a) The results are very good across several benchmarks and settings
b) It appears that the method can be applied in any other similar settings while preserving some privacy

Cons \
a) The first half of the paper that describes the methods is rather confusing; too much information that is hard to follow through - figure 2 does not have a caption and there are a couple of references to figures in the text that are not correct. \
b) Even after reading the sections a few times I struggle to find how endo- and exo-temporal regularisations are defined; yes some equations do exist etc. but the clarity is poor. Better structured information is required to comprehend the significance and contextualise the results with respect to the methods.
Can you please elaborate on this? "Overall, the proposed endo-temporal regularization drives the target model to obtain discriminable temporal features by extracting clip features with higher discriminability that complies with the masked-temporal hypothesis and the cluster assumption." How does it push it to obtain discriminable features?

---

> ### Comment · Reviewer_pFxC · 2023-12-16
> **Responses**
>
> I would appreciate it if the authors provided some responses at their earlier convenience

---

> ### Author Response · Authors · 2023-12-16
> **Response to Reviewer pFxC (1/2)**
>
> We sincerely apologize for the delay in response to Reviewer pFxC. We sincerely thank the reviewer for his thorough summarization of our work and acknowledging its contributions. We are also glad that the reviewer acknowledges the applicability and performance of our proposed method. Here we answer all the questions and concerns raised by the reviewer. We hope the responses can address the concerns and conform to the requested changes. In addition, we have submitted a revised manuscript we highlighted all the updates in blue color.
>
> **Q1:**
> > The first half of the paper that describes the methods is rather confusing; too much information that is hard to follow through - figure 2 does not have a caption and there are a couple of references to figures in the text that are not correct.
>
> **Answer:** We apologize for the confusion and poor readability over the description of the method. We have revamped the description to improve the readability. Specifically, we explain the definition of the endo-temporal and exo-temporal regularizations separately and concisely in Sec. 1 on Page 2-3, quoted as (and highlighted in blue in the updated manuscript):
> > Specifically, the endo-temporal regularization is designed to improve the discriminability of clip features by regularizing the behavior of clip features obtained __within the same target video__.
>
> > Meanwhile, the exo-temporal regularization is designed to drive the proposed model to extract more stable and discriminable temporal features, characterized by being linearly smooth in-between features obtained from __different target videos__.”
>
> The definitions of both regularizations have also been reiterated when they are described in detail in Sec. 3.2 on Page 5-6. Meanwhile, we have added the missing caption for Figure 2, and checked the manuscript to ensure the references to the different figures are correct.
>
> **Q2:**
> > Even after reading the sections a few times I struggle to find how endo- and exo-temporal regularisations are defined; yes some equations do exist etc. but the clarity is poor. Better structured information is required to comprehend the significance and contextualise the results with respect to the methods.
>
> **Answer:** We apologize for the poor clarity over how endo- and exo-temporal regularizations are defined. As mentioned in the **Answer** for **Q1** of Reviewer pFxC, we have revised the introduction and given clearer explanations to the definition of the two regularizations separately and concisely, in Sec. 1 on Page 2-3. We have also reiterated the definitions in Sec. 3.2. Meanwhile, we have also re-organized the structure of Sec. 3.2 with the addition of two separate paragraphs. Each paragraph corresponds to the endo-temporal regularization and the exo-temporal regularization respectively.
>
> **Q3:**
> > Can you please elaborate on this? "Overall, the proposed endo-temporal regularization drives the target model to obtain discriminable temporal features by extracting clip features with higher discriminability that complies with the masked-temporal hypothesis and the cluster assumption." How does it push it to obtain discriminable features?
>
> **Answer:** As mentioned in Sec. 1, the _masked-temporal hypothesis_ which matches the human intuition essentially states that if the obtained target clip features enables the target model to obtain consistent predictions with only partial clip information, the corresponding temporal features would possess high discriminability.
>
> This is because such clip features would tend to gather close around the corresponding temporal features and would not scatter across decision boundaries as depicted in the right-most figure of Figure 1. Alternatively, for clip features that are scattered across decision boundaries, there are chances where the virtual temporal features would not produce the same label prediction as the temporal features, as displayed in the middle figure of Figure 1. Meanwhile, this simultaneously requires the clip features to comply with the cluster assumptions.
>
> Therefore, optimizing the target model such that the clip features comply with both the masked-temporal hypothesis and the cluster assumption would result in more discriminable temporal features. To comply with both criteria, the endo-temporal regularization is applied to regularize the behavior of clip features from the same target video, achieved via the optimization of the loss function $\mathcal{L}_{endo}$ (as in Eq. 7).
>
> Thus optimizing the loss function $\mathcal{L}_{endo}$ would ultimately push the target model towards obtaining more discriminable temporal features.

---

> > ### Comment · Reviewer_pFxC · 2023-12-19
> >
> > Thanks for the responses to my questions - I read the revised version of the paper and it has been considerably improved. The novelty, albeit limited, has been better fleshed out and the explanations on the regularisations are clearer.

---

> ### Author Response · Authors · 2023-12-16
> **Response to Reviewer pFxC (2/2)**
>
> **Q4:**
> > Clarify needs to be improved - the whole paper seems to be contextualized around the two regularization terms so this needs to be fleshed out more clearly.
>
> **Answer:** We gratefully thank the reviewer for his suggestion and acknowledge that the definition of the two regularization terms (endo-temporal and exo-temporal regularizations) are not clear enough. We have therefore made efforts to clarify the two regularization terms by clearer definition and restructured the description of the regularization terms, as mentioned in our **Answer**s to **Q1** and **Q2**.
>
> **Q5:**
> > Figure 2 needs fixing.
>
> **Answer:** We sincerely apologize the mistakes made in Figure 2 which hampered the readability of the manuscript. We have fixed Figure 2 as mentioned in **Answer** to **Q1**to improve the readability and have add in the missing caption.
>
> **Q6:**
> > There are several typos and long sentences.
>
> **Answer:** We thank the reviewer for his checking and apologize for the typos and long sentences. We have gone through the manuscript carefully and have corrected the typos and sentences. We have also made effort in improving the readability of the manuscript. The modifications of the manuscript are highlighted in blue.
>
> **Q7:**
> > Conclusion is short; please expand and provide a short discussion on importance, generalization and future perspectives.
>
> **Answer:** We agree with the reviewer that the previous conclusion is short and thank the reviewer for his suggestions on expanding the conclusion. We have thus expanded the conclusion section with discussions on the importance of EXTERN as well as the future perspectives for improving EXTERN, quoted as (and highlighted in blue on Pages 13 - 14):
>
> >We believe that such a superior performance of EXTERN could pave a new way for tackling video domain adaptation without compromising data privacy.
>
> > While the proposed EXTERN has proven to be effective for the BVDA task, there is still room for future improvements. We observe that EXTERN may not perform well in tasks where the domain gap is relatively large, as depicted in Table 2 for tasks involving the ARID domain. This suggest that the current endo- and exo-temporal regularizations which are built on features constructed with the linear MixUp approach may not work under large domain shift scenarios. Virtual temporal features and interpolated temporal features constructed via higher order computation could be explored. Further, the temporal relationship between clip features could also be leveraged more explicitly as an additional modality for adaptation. Additionally, current BVDA tasks have only been explored for the closed-set and partial-set scenarios. More challenging settings such as the open-set setting where the target video domain may contain target-private classes may also be further explored.

---

### Review · Reviewer_SMeM · 2023-11-26

**Summary Of Contributions:**

This paper discusses black box video domain adaptation (BVDA) for privacy protection and portable video model transfer. Although several black-box region adaptation methods have been proposed for the image domain, these methods are not applicable to the video domain because the video modality has more complex temporal features and is more difficult to align. To address BVDA, this paper proposes a new Endo and eXo-TEmporal Regularized Network (EXTERN) by applying mask-to-mix strategies and video-adapted regularization. The proposed method drives clip features to satisfy the masked-temporal hypothesis and the cluster assumption to obtain practical and discriminative temporal features. Empirical results show that EXTERN exhibits high adaptive performance compared to the baseline methods on various cross-domain closed-set and partial-set action recognition benchmarks.

**Audience:**

Yes

**Claims And Evidence:**

Yes

**Requested Changes:**

The paper needs more elaboration, and the reviewer thinks it is necessary to clarify some of the concerns mentioned in the Weaknesses section.

**Strengths And Weaknesses:**

Strengths
- This paper is one of the first studies to address black box video domain adaptation (BVDA).
- This paper proposes a novel pipeline, EXTERN, to deal with BVDA.
- This paper reveals that EXTERN exhibits high adaptation performance compared to the baseline methods on various cross-domain closed-set and partial-set action recognition benchmarks.
- The proposed method is simple, and the paper is easy to read.

Weaknesses
- The paper mentions complex temporal features as a difficulty of Video Domain Adaptation (VDA), but only the temporal generator G_t explicitly deals with temporal information. After G_t, clip features are mainly used, but the temporal relationship between clip features is rarely considered.
- Motivation to use MixUp could be stronger. At the very least, the motivation or necessity should be stated more clearly in the paper than it is now. For example, the masked temporal hypothesis states that mixup features are learned to approach temporal features. However, it is equally effective if the clip features are learned to approach the temporal features instead of the mixup features directly.
- The core technology of the proposed method relies heavily on interpolation consistent training (ICT) and MixUp, so the technical novelty is not that high.
- As a paper, it needs more elaboration. For example, the figure on page 4 has no caption. Symbols such as \mathcal{V}, \mathcal{Y}, and \mathcal{H} are used without definitions. Also, in Equation 8, what does it mean to compute the cross entropy loss between temporal feature t and the prediction result?
- Theoretical background on the proposed method is lacking.

---

> ### Author Response · Authors · 2023-12-16
> **Delayed Responses to Reviewer SMeM**
>
> Dear Reviewer SMeM:
>
> We sincerely thank you for your great effort made in reviewing our work and the various suggestions proposed. We would like to sincerely apologize for our delayed responses to you as we are striving to address all concerns and suggestions with further updates on the manuscript. We are aiming for the completion of our updated manuscript and responses to your concerns by next Monday, Dec. 18, 2023. We would like to seek your kind understanding over the delays.
>
> Thank you!
>
> Warmest Regards,
>
> Authors of TMLR Paper 1612

---

> ### Author Response · Authors · 2023-12-17
> **Response to Reviewer SMeM (1/2)**
>
> We once again sincerely apologize for the delayed response to Reviewer SMeM. We gratefully thank the reviewer for his thorough summarization of our work. We also sincerely thank the reviewer for his recognition over our proposed method, which is indeed simple yet effective for BVDA. We further thank the reviewer for his appreciation in the readability of our manuscript. Here we answer all the questions and concerns raised by the reviewer. We hope the responses can address the concerns and conform to the requested changes. A revised manuscript have also been submitted where we highlighted all the updates in blue color.
>
> **Q1:**
>
> > The paper mentions complex temporal features as a difficulty of Video Domain Adaptation (VDA), but only the temporal generator G_t explicitly deals with temporal information. After G_t, clip features are mainly used, but the temporal relationship between clip features is rarely considered.
>
> **Answer:** We agree with the reviewer that the current method could be further improved since our current method leverages the clip features mainly through the construction of virtual temporal features and the application of the endo-temporal regularization term. We thank the reviewer for his suggestion that a more comprehensive temporal relationship between the clip features could be considered as an additional modality for adaptation. We have included this as a potential future perspective in Sec. 5 on Page 14, quoted as follows:
>
> > While the proposed EXTERN has proven to be effective for the BVDA task, there is still room for future improvements. We observe that EXTERN may not perform well in tasks where the domain gap is relatively large, as depicted in Table 2 for tasks involving the ARID domain. This suggest that the current endo- and exo-temporal regularizations which are built on features constructed with the linear MixUp approach may not work under large domain shift scenarios. Virtual temporal features and interpolated temporal features constructed via higher order computation could be explored. Further, the temporal relationship between clip features could also be leveraged more explicitly as an additional modality for adaptation.
>
> **Q2:**
>
> > The masked temporal hypothesis states that mixup features are learned to approach temporal features. However, it is equally effective if the clip features are learned to approach the temporal features instead of the mixup features directly.
>
> **Answer:** We thank the reviewer for the concern raised. We have validated the necessity of complying with the *masked-temporal hypothesis* (and thus the optimization of $\mathcal{L}_ {pre}$) in the Ablation studies of Table 4 by comparing the last row with the sixth row. We observe a notable performance drop when the *masked-temporal hypothesis* is not considered hence the $\mathcal{L}_ {pre}$ loss is not optimized.
>
> Further, we agree that we should design a further experiment to validate the effectiveness of leveraging mixed clip features instead of directly push individual clip features towards the temporal features. We denote the variant of EXTERN where the loss $\mathcal{L}_ {pre}$ is formulated by the average KL divergence between the target prediction of all clip features and the temporal feature as EXTERN-direct. The performances of EXTERN-direct are compared with EXTERN as follows:
>
> | Method | U101$\to$H51 | H51$\to$U101 | U-14$\to$H-7 | H-14$\to$U-7 | Average |
> | :---------- | :------------------: | :------------------: | :----------------: | :----------------: |:----------: |
> | EXTERN-direct | 86.80 | 90.63 | 70.04 | 89.22 | 84.17 |
> | EXTERN | 88.89 | 91.95 | 71.43 | 90.60 | __85.72__ |
>
> It can be observed that the performances of EXTERN-direct are notably inferior than that of our proposed EXTERN. This empirically prove that leveraging a mix of randomly unmasked clip features to comply with the *masked-temporal hypothesis* is more effective than directly push clip features to learn to approach the temporal features.

---

> ### Author Response · Authors · 2023-12-17
> **Response to Reviewer SMeM (2/2)**
>
> **Q3:**
>
> > The core technology of the proposed method relies heavily on interpolation consistent training (ICT) and MixUp, so the technical novelty is not that high.
>
> **Answer:** We agree with the reviewer that the technical novelty of our proposed EXTERN is not hugely significant. Instead, in this paper, the key novelty is the proposal of a new and more realistic video domain adaptation task BVDA that protects both the source data and source model privacy that results in more portable video models. To address the newly proposed BVDA task without source data and model access, we resort to an alternative strategy where we push clip features and temporal features towards higher discriminability, thus formulate the _masked-temporal hypothesis_. We empirically show by obtaining clip features and temporal features that comply with the proposed hypothesis, the resulting target model achieved excellent performances on the BVDA tasks under both closed-set and partial-set settings. Inspired by prior studies on the relevancy between linear in-between feature behavior and feature discriminability, we resort to MixUp and ICT to obtain features that comply with the _masked-temporal hypothesis_.
>
> **Q4:**
>
> > As a paper, it needs more elaboration. For example, the figure on page 4 has no caption. Symbols such as \mathcal{V}, \mathcal{Y} are used without definitions. Also, in Equation 8, what does it mean to compute the cross entropy loss between temporal feature t and the prediction result?
>
> **Answer:** We apologize for the lack of elaboration and mistakes made in our equations. The missing caption of Fig. 2 on Page 4 has been supplemented, highlighted in blue. We have also included the definitions of the symbols \mathcal{V} and \mathcal{Y} at the beginning of Sec. 3 on Page 4, quoted as follows and highlighted in blue:
> > $\mathcal{V}_ {S}$ is the source input distribution and $\mathcal{Y}_ {S}$ the source label space. …, where $\mathcal{V}_ {T}$ is the target input distribution. …, $\mathcal{Y}_ {S}=\mathcal{Y}_ {T}$ with $\mathcal{Y}_ {T}$ denoting the target label space.”
>
> There is a mistake in the previous Eq. 8, where the target predictor $H_ {T}$ is missing. Eq. 8 has been updated in Sec. 3.2 on Page 7 highlighted in blue as:
> > $\mathcal{L}_ {exo} = l_ {ce}(H_ {T}(\mathrm{MixUp}_ {\lambda_{t}} (\mathbf{t}_ {i}, \mathbf{t}_ {j})),\, \mathrm{MixUp}_ {\lambda_{t}} (y_ {i}, y_ {j}))$.
>
> **Q5:**
>
> > Theoretical background on the proposed method is lacking.
>
> **Answer:** The current paper focuses more on the proposal of a novel and more realistic video adaptation task, while proposing a simple yet effective EXTERN to address BVDA. The effectiveness of EXTERN is validated through extensive empirical study instead of theoretical analysis. We would like to admit that there is no theoretical contribution towards the learning theory of domain adaptation in this paper.
>
> Despite the lack of theoretical analysis, we have empirically demonstrated that our proposed EXTERN outperforms existing black-box adaptation (BDA) methods and even several current video unsupervised domain adaptation (VUDA) methods with access to source data for the BVDA task under both closed-set and and partial set video domain adaptation benchmarks.
>
> We have considered the possibility of tacking BVDA with inspiration from prior theoretical-inspired methods, such as the discrepancy-based methods MK-MMD and ACAN, or adversarial-based methods DANN and SAVA. However, these methods require source data access which violates the data-private setting of BVDA. Meanwhile, the superior performance of EXTERN suggest that  training a target model from scratch with strong regularizations while adapting solely with source predictions can be as effective as data-based domain alignment techniques. Yet, we agree that further exploration over how our proposed EXTERN contribute further to the domain adaptation learning theory would be necessary in the future development of BVDA and would like to thank the reviewer for his comment.
>
> **Q6:**
>
> > The paper needs more elaboration, and the reviewer thinks it is necessary to clarify some of the concerns mentioned in the Weaknesses section.
>
> **Answer:** We thank the reviewer for his suggestion and agree with his concerns mentioned above. We have made efforts in addressing the concerns by supplementing the missing captions, clarifying the definitions of symbols and proposed terms, and by elaborating on current motivation and future works as above. We hope our update could clarify the concerns raised above.

---

### Review · Reviewer_Qj8M · 2023-12-11

**Summary Of Contributions:**

This work proposes and addresses a novel Black-box Video Domain Adaptation (BVDA) task with a proposed EXTERN model. Based on the masked-temporal hypothesis and cluster assumption, the authors propose two forms of regularizations. The endo-temporal regularization term is based on the target temporal feature, virtual temporal feature generated with mask-to-mix strategy and MixUp predictions. The exo-temporal regularization term is calculated by ICT across corresponding temporal features. The whole network is based on the two regularization terms with knowledge distillation operations, clip feature weights and information maximization loss. Experiments comprehensively evaluate the effectiveness of the proposed EXTERN method.

**Audience:**

Yes

**Broader Impact Concerns:**

None.

**Claims And Evidence:**

Yes

**Requested Changes:**

Please refer to the Weaknesses above.
Further, some domain adaptation reviews and surveys are suggested, such as “Source-free unsupervised domain adaptation: A survey”, “A Comprehensive Survey on Source-free Domain Adaptation”, “A Review of Single-Source Deep Unsupervised Visual Domain Adaptation”. Please also add a caption for Figure 2.

**Strengths And Weaknesses:**

Strengths: One of the main contributions, the two proposed regularization terms are built with random masking and MixUp operations, making it easy to comprehend and reproduce. Meanwhile, the proposed method with these regularization terms obtains superior performance against VUDA and SFVDA approaches in most experiments. This also proves the correctness of the proposed masked-temporal hypothesis. This hypothesis will not only promote future work about BVDA but also be a substantial inspiration for VUDA and SFVDA tasks. The quality, clarity, and significance of this work are outstanding.

Weaknesses: As an important concept in the paper, cluster assumption does not get any definition elaboration. I suggest you give a brief definition and briefly explain why the initial features violate the cluster assumption in Figure 1. Furthermore, you assign the same weight to the two regularization terms. I think you can conduct more experiments exploring the relative importance of the two regularization terms individually if possible, as they may have distinct contributions to the final performance.

---

> ### Author Response · Authors · 2023-12-16
> **Response to Reviewer Qj8M (1/2)**
>
> We sincerely thank the reviewer for his thorough summarization and his comprehensive understanding of our work. In addition, we gratefully thank the reviewer for his recognition over the importance and potential of BVDA and our proposed method and hypothesis. We thank the reviewer for his great appreciation of our work. We also thank the reviewer for his subsequent suggestions and have made amendments and updates accordingly.
>
> **Q1:**
> > As an important concept in the paper, cluster assumption does not get any definition elaboration. I suggest you give a brief definition and briefly explain why the initial features violate the cluster assumption in Figure 1.
>
> **Answer:** We thank the reviewer for his suggestion and agree that the cluster assumption should be mentioned and defined more clearly, while the captions of Fig. 1 should be also more explanatory. We have therefore elaborated on the definition of the cluster assumption in Sec. 3 Page 4 quoted as (and highlighted in blue in the updated manuscript):
>
> > Essentially, for BVDA, such a strategy aims to extract effective temporal features with high discriminability and complies with the cluster assumption, where the input distribution contains separated data clusters and that data samples in the same cluster share the same class label.
>
> We have also modified the caption of Fig. 1 on Page 2 to briefly explain why the initial features violate the cluster assumption quoted as:
>
> > Clip features of target videos may be scattered, where clips from the same video may be separated by the decision boundary, resulting in differed label predictions. Such clip features violate both the cluster assumption and the _masked temporal hypothesis_.
>
> **Q2:**
> > Furthermore, you assign the same weight to the two regularization terms. I think you can conduct more experiments exploring the relative importance of the two regularization terms individually if possible, as they may have distinct contributions to the final performance.
>
> **Answer:** We sincerely thank the reviewer for pointing out his observation and for his suggestion in further exploration on the relative importance of the two regularization terms. We agree with the observation that the two regularization terms may have distinct contributions since it is observed that the drop in EXTERN’s performance varied when the two regularization terms are applied individually. We therefore explored how the different regularization weights (denoted as $\beta_{reg}^{ex}$ for the exo-temporal regularization and $\beta_{reg}^{en}$ for the endo-temporal regularization) affect the performance of EXTERN. The results are as follows and also displayed in Table 5 Page 11 of the manuscript. Note that the following table is simplified due to formatting issues.
>
> | $\beta_{reg}^{ex}$ | $\beta_{reg}^{en}$ | U101$\to$H51 | H51$\to$U101 | U-14$\to$H-7 | H-14$\to$U-7 | Average |
> | :-----------------------: | :-----------------------: | :------------------: | :------------------: | :----------------: | :----------------: |:----------: |
> | 1.0 | 1.0 | 88.89 | 91.95 | 71.43 | 90.60 | __85.72__ |
> | 1.0 | 0.0 | 80.08 | 86.34 | 61.91 | 82.71 | 77.76 |
> | 1.0 | 0.1 | 82.75 | 88.49 | 64.57 | 84.34 | 80.04 |
> | 1.0 | 0.5 | 86.35 | 91.59 | 69.02 | 88.20 | 83.79 |
> | 1.0 | 1.5 | 88.53 | __92.12__ | __72.48__ | 89.42 | 85.64 |
> | 1.0 | 2.0 | __89.28__ | 91.42 | 70.80 | __90.84__ | 85.59 |
> | 0.0 | 1.0 | 85.83 | 90.37 | 67.71 | 88.91 | 83.21 |
> | 0.1 | 1.0 | 86.44 | 90.84 | 68.45 | 89.33 | 83.76 |
> | 0.5 | 1.0 | 88.43 | 91.48 | 70.98 | 90.31 | 85.30 |
> | 1.5 | 1.0 | 88.24 | 90.84 | 71.75 | 90.18 | 85.25 |
> | 2.0 | 1.0 | 87.92 | 91.10 | 70.42 | 89.88 | 84.83 |
>
> The results suggests that the endo-temporal regularization would contribute more towards EXTERN's performance. It is further observed that when $\beta_{reg}^{ex}=1.0$ and $\beta_{reg}^{en}\leqslant 1.0$, a small weight increase in the endo-temporal regularization would result in notable performance boost. Meanwhile, EXTERN achieves the overall best performance when both regularizations are balanced, which demonstrates the correctness of our chosen hyperparameter setting. A more detailed analysis is presented in Sec. 4.3 on Page 12.

---

> ### Author Response · Authors · 2023-12-16
> **Response to Reviewer Qj8M (2/2)**
>
> **Q3:**
> > Please refer to the Weaknesses above. Further, some domain adaptation reviews and surveys are suggested, such as “Source-free unsupervised domain adaptation: A survey”, “A Comprehensive Survey on Source-free Domain Adaptation”, “A Review of Single-Source Deep Unsupervised Visual Domain Adaptation”. Please also add a caption for Figure 2.
>
> **Answer:** We thank the reviewer for his suggestions and apologize for the missing references and caption. The reviews and surveys are of relevance to our paper and we have included all in our updated Related Work section. We have also add in the caption for Fig. 2 on Page 4 quoted as (and highlighted in blue in the updated manuscript):
>
> > An overview of the proposed EXTERN. EXTERN extracts knowledge from the black-box source predictor (i.e., Source API) through a distillation process. EXTERN further extracts temporal features in a self-supervised manner by applying both the _endo-temporal regularization_ and _exo-temporal regularization_. To apply the _endo-temporal regularization_, the virtual temporal features are constructed with a mask-to-mix strategy.

---

### Decision · Action_Editor_3hbg · 2024-01-29

**Recommendation:** Accept as is

**Comment:**

Three expert reviewers raised some concerns about the paper that were resolved by the authors' response. Post-response, all reviewers were in favor of accepting the paper, indicating that the paper was novel and clearly demonstrated its claims. The AE is in agreement and recommends the paper for acceptance.

**Audience:**

Yes, the reviewers think that people in TMLR's audience would be find this paper of interest. The AE is in agreement.

**Claims And Evidence:**

Yes. All three reviewers think that the revised version of the paper is supported by clear evidence. The AE is in agreement.

---

> ### Author Response · Authors · 2024-02-15
> **Appreciation of Acceptance with Camera Ready Version Uploaded**
>
> Dear Action Editor 3hbg:
>
> We sincerely thank your effort and also the efforts of all reviewers throughout the reviewing process. The reviews have helped in improving this article significantly. We have uploaded the Camera Ready Version as instructed.
>
> Thank you for your time and effort!
>
> Warmest Regards,
>
> Authors of TMLR Paper 1612